# Spread Through Air Spaces (STAS) as a Predictive and Prognostic Factor in Patients with Non-Small Cell Lung Cancer—Systematic Review

**DOI:** 10.3390/cancers17101696

**Published:** 2025-05-18

**Authors:** Mikołaj Herba, Sylwia Boczek, Wiktoria Smyła-Gruca, Katarzyna Kost, Damian Czyżewski, Mateusz Rydel

**Affiliations:** 1Faculty of Medical Sciences in Zabrze, Student Scientific Society at Department of Thoracic Surgery, Medical University of Silesia, 40-055 Katowice, Poland; sylwiaboczek10@gmail.com (S.B.); wiktoriasmyla@gmail.com (W.S.-G.); katarzynaagorny@gmail.com (K.K.); 2Faculty of Medical Sciences in Zabrze, Department of Thoracic Surgery, Medical University of Silesia, 40-055 Katowice, Poland; dczyzewski@sum.edu.pl (D.C.); mateusz.rydel@wp.pl (M.R.)

**Keywords:** STAS, lung cancer, thoracosurgery

## Abstract

Lung cancer is the second most prevalent cancer in the general population and the third most prevalent among women. STAS (Spread Through Air Spaces) is a term used in pathology, particularly in lung cancer. It refers to the spread of tumor cells through air spaces in the lung tissue. The available literature was analyzed to assess the possibility of using the presence of STAS as a predictive and prognostic factor in all treatment methods for non-small cell lung cancer, as well as the effectiveness of various imaging studies in assessing the probability of STAS occurrence.

## 1. Introduction

Lung cancer is the second most common form of cancer in the general population, the first most common among men, and the third most common among women (Figure 1). It also has the highest mortality rate of all cancer types (Figure 2) [1]. Globally, in 2020, there were approximately 2,206,771 new cases of lung cancer, including 1,435,943 cases in males and 770,828 in females. Among male patients, 923,237 cases (89%) were non-small cell lung cancer (NSCLC), which included 560,108 cases (39%) of adenocarcinoma (ADC), 351,807 cases (25%) of squamous cell carcinoma (SCC), and 115,322 cases (8%) of large cell carcinoma (LCC). Small cell lung cancer (SCLC) accounted for 163,862 cases (11%). In female patients, NSCLC occurred in 580,827 cases (91%), with adenocarcinoma accounting for 440,510 cases (57%), squamous cell carcinoma for 91,070 cases (12%), and large-cell carcinoma for 49,246 cases (6%). Small-cell carcinoma was observed in 68,224 cases (9%). Eastern Asia had the highest age-standardized incidence rates per 100,000 person-years for adenocarcinoma, squamous cell carcinoma, small-cell carcinoma, and large-cell carcinoma, with rates of 23.5 for males and 16.0 for females [2]. Bing-Yen Wang et al. demonstrated that ADC patients exhibit significantly higher stage-specific 5-year overall survival rates across all clinical stages, as shown in both univariate and multivariate analyses [3].

Lung cancer remains one of the most challenging malignancies, with early detection and effective management being paramount. In this context, tumor markers have emerged as vital tools in both the diagnosis and prognostication of lung cancer.

Some biomarkers serve as prognostic factors in NSCLC. Vascular endothelial growth factor (VEGF), the presence of which is associated with the inherence of metastases, tumor recurrence, decreased survival, and increased risk of death [4,5,6]. HIF-1α is a modulator of the immune response and is involved in the mechanism of cisplatin resistance in NSCLC [7,8]. High levels of YKL-40 are correlated with shorter survival in cancer patients, which may be related to its role in angiogenesis [9,10].

STAS is a novel concept in pulmonary adenocarcinomas introduced in the 2015 World Health Organization (WHO) classification for lung cancer. The WHO defined STAS as “micropapillary clusters, solid nests, or single cells spreading within air spaces beyond the edge of the main tumor” back in 2015. This definition added STAS as an additional criterion for adenocarcinoma invasion, supplementing the existing criteria (invasion into myofibroblastic stroma, vascular or pleural invasion, and histological subtypes other than the lepidic pattern) [11]. Patients with solid (SOL) and/or micropapillary (MIP) histologic subtypes, as well as those with tumors exhibiting STAS, are at an increased risk of recurrence. Unlike conventional modes of tumor invasion, such as pleural, stromal, and lymphovascular invasion, pulmonary metastases arising from STAS do not adhere to the typical tumor margins within normal lung tissue. This unfavorable prognostic factor, identified as STAS, has also been associated with locoregional recurrence following limited resection (LR) in ADC and other lung cancer types, including SCC, pleomorphic carcinoma, and neuroendocrine neoplasms [12].

Numerous studies have demonstrated that STAS serves as an independent prognostic factor [4,13,14]. The presence of STAS has been identified as a risk factor for occult nodal metastases. Moreover, it has also been associated with lymphatic, vascular, and pleural invasion [5].

Before the concept of STAS was introduced, sublobar resection was considered an acceptable treatment option for tumors ≤ 3 cm in diameter. However, recent findings suggest that the presence of STAS significantly increases the risk of recurrence and worsens the prognosis when sublobectomy is performed instead of lobectomy [6,7,15]. Therefore, evaluating STAS status has become essential in determining the appropriate extent of surgical resection.

Currently, the gold standard for STAS diagnosis is postoperative pathological examination. Unfortunately, this approach limits the ability to make preoperative decisions about the surgical strategy. As a result, ongoing research aims to develop preoperative diagnostic methods that can accurately identify the presence of STAS and support optimal surgical planning [6].

Preoperative assessments using bronchoscopic cytology and intraoperative frozen section analysis have been attempted, but their sensitivity and accuracy remain insufficient. Better outcomes have been achieved by integrating histopathological evaluation with clinical data and imaging studies [6,16].

Imaging techniques are also being explored as potential diagnostic tools. Computed tomography (CT) can reveal features suggestive of STAS, but this method relies heavily on the radiologist’s subjective interpretation, increasing the risk of diagnostic errors. To address this, radiomics is being investigated as a more objective alternative. Radiomics involves converting imaging data into quantifiable features that can be analyzed to inform clinical decisions. Importantly, it is recommended to evaluate not only the tumor itself but also the peritumoral area. This region may be more informative in the context of STAS, given that the condition involves tumor cell spread beyond the primary tumor margin. Although research on the peritumoral area remains limited, available studies indicate its potential utility in preoperative STAS assessment. Ultimately, there is a need to develop a robust risk stratification model that combines clinical data with intratumoral and peritumoral imaging features to accurately predict STAS presence. This is particularly important for patients with tumors ≤ 3 cm, for whom accurate assessment is critical in determining the extent of surgical resection [6].

The identification of STAS in NSCLC affects treatment decisions. However, data regarding the use of chemotherapy, radiotherapy, and immunotherapy in patients with STAS are limited [5,17,18,19,20,21].There are various research suggesting that the presence of STAS may be crucial when surgeons decide on the extent of lung resection in surgical treatment of NSCLC [18,22,23,24,25,26,27,28,29].

The aim of this study was to evaluate the utility of STAS as a predictive and prognostic factor, as well as to assess the impact of STAS detection on subsequent surgical and pharmacological treatment decisions. Additionally, a literature review was conducted to deepen understanding of STAS, which remains an active area of research and is often underrecognized as a clinically significant factor.

## 2. Material and Methods

To support our claims, we conducted a literature search using PubMed, PMC, and Google Scholar from June to September 2024. The search terms included phrases such as: “STAS”, “lung cancer”, “NSCLC”, “SCLC”, “PET and STAS”, “histopathological STAS”, “treatment methods for STAS”, and “STAS prognosis”. Initially, titles and abstracts were screened to identify relevant studies. To achieve a broader understanding of the topic and include substantial research data, we incorporated various study types, including meta-analyses, case-control studies, literature reviews, cross-sectional studies, and prospective, longitudinal studies.

The inclusion criterion for selecting publications addressing the topic of STAS was a publication date from 2015 onwards. We limited our search to studies focusing on STAS in non-small cell lung cancer; studies exclusively concerning small cell lung cancer were excluded. The analysis of the included publications was conducted independently by four authors. The databases were searched independently as well.

As our work is classified as a systematic review, we did not perform statistical analyses on the data we collected. Furthermore, a significant limiting factor of our systematic review was the limited number of available studies addressing the topic of STAS. It was challenging to establish unequivocal inclusion criteria, as the existing literature employed widely varying definitions and scopes of STAS. Instead, we compiled the available information and presented it descriptively. The limited number and overall volume of studies underscore that STAS remains an emerging area of research, warranting further investigation.

Considering our study as a systematic review, ethical approval by a bioethics committee was not required.

## 3. Results

### 3.1. STAS in Different Histological Subtypes of NSCLC

As noted earlier, NSCLC is classified into three main subtypes: ADC, SCC, and LCC. The prevalence of STAS varies across these subtypes. STAS is most commonly observed in ADC, with reported rates ranging from 28.2% to 51.4%. In LCC, STAS occurrence is estimated at approximately 43%, while in SCC, the prevalence ranges from 19.1% to 40.3% [30].

Advanced age (>65 years), male sex, smoking history, and abnormal serum carcinoembryonic antigen level have been identified as clinical factors associated with a higher prevalence of STAS in ADC. Both ADC and SCC share several pathologic parameters linked to STAS positivity, including larger tumor size, lymphovascular invasion, and pathological stages. Lee et al. conducted a retrospective analysis of 316 surgically resected lung ADC cases, providing valuable insights into these associations [31].

They concluded that STAS was independently associated with shorter RFS and was linked to recurrences at both extrathoracic and intrathoracic sites [31]. Similarly, Liu et al. found that patients with high expression of metastasis-associated protein 1 (MTA1) and STAS positivity had significantly worse OS and shorter RFS compared to others (*p* < 0.001) [32]. This study focused on patients with stage I–III lung adenocarcinoma [32]. Overall, the findings for ADC are in accordance with findings reported SCC. Kadota et al. demonstrated that among patients with SCC, 5-year RFS was significantly lower for those with STAS compared to those without STAS (*p* = 0.001) [33]. In contrast, data from Yanagawa et al. indicated that while STAS was associated with recurrence and worse survival in stage I SCC, no such association was observed in stage II and III squamous cell carcinoma [8]. Meta-analyses provide a broader perspective on individual studies. A 2019 meta-analysis including 3754 patients with NSCLC found that the presence of STAS was significantly associated with worse RFS (HR, 1.975; 95% CI, 1.691–2.307; *p* < 0.001) and overall survival (OS) (HR, 1.75; 95% CI, 1.375–2.227; *p* < 0.001) [34]. A similar conclusion was reached by another meta-analysis, which points out that positive STAS might be an unfavorable prognostic factor for patients with NSCLC. A third large meta-analysis by Wang et al. further supports this conclusion, suggesting that STAS serves as a novel prognostic predictor in NSCLC [3]. In this study, which included 3231 NSCLC patients, STAS was observed in 1204 cases (37.3%). The presence of STAS was strongly associated with poor PFS ([HR], 1.789; *p* < 0.001) and OS (HR, 1.488; *p* < 0.001) [3].

### 3.2. Intraoperative Discussion

The evaluation of STAS using intraoperative frozen sections is becoming increasingly popular. This technique was developed to assist surgeons in deciding whether to perform a lobectomy or a sublobar resection for patients with lung ADC. Eguchi et al. provided evidence supporting the utility of this approach [7]. In their study, 1497 patients with T1N0M0 lung ADC were analyzed: 970 underwent lobectomies, while 537 underwent sublobar resections. Among patients who underwent sublobar resection, recurrence patterns (both locoregional and distant) were evaluated in relation to the margin- to-tumor ratio. Multivariable analysis revealed that sublobar resection was significantly associated with lung cancer-specific mortality (subhazard ratio: 2.63; *p* = 0.021) and recurrence (subhazard ratio: 2.84; *p* < 0.001) in patients with STAS. The sensitivity and specificity of frozen section for detecting STAS were 71% and 92%, respectively. In contrast, findings by Zhou et al. showed a different perspective [9]. Their retrospective study independently assessed the presence of STAS using frozen and permanent sections. For STAS detection, frozen sections demonstrated a lower sensitivity (55%) and positive predictive value (48%) compared to permanent sections, along with moderate agreement (K = 0.34). However, specificity (80%) and negative predictive value (85%) were relatively higher. The authors cautioned that relying on STAS detection in frozen sections as a threshold for lobectomy could lead to overdiagnosis, resulting in 13 unnecessary lobectomies (or 8% of the 163 cases). To make STAS detection a more reliable tool in routine surgical practice, future efforts should focus on identifying histologic markers that can differentiate STAS with sufficient sensitivity and specificity during frozen section analysis [9].

### 3.3. Imaging Diagnostics

Research is ongoing to determine whether preoperative evaluation of STAS can be achieved using diagnostic imaging techniques, such as CT and PET/CT. Several imaging factors have been identified that may help predict the presence of STAS. According to current data, pure-solid nodules, part-solid nodules with large solid components, and lesions with a high maximal standardized uptake value (SUVmax) are more likely to be STAS-positive and should be managed with lobectomy rather than limited resection. Conversely, pure ground-glass nodules and subsolid nodules with a predominant ground-glass component are typically STAS-negative and may be suitable for limited resection [35]. When evaluating frozen sections, pathologists could enhance STAS detection by focusing on cases with CT morphological or quantitative features associated with STAS, which may warrant more extensive frozen section sampling. Future research should further explore these imaging and morphological factors, as they could help reduce diagnostic uncertainty in the preoperative assessment of STAS [36].

#### 3.3.1. Positron Emission Tomography (PET)

Nishimori et al. investigated the potential of using 18F FDG-PET/CT to predict the presence of STAS in clinical stage I lung ADC [10]. They calculated and compared three PET parameters—SUV max, metabolic tumor volume (MTV), and total lesion glycolysis (TLG)—between two patient groups: those with and without STAS confirmed during pathological testing. Their findings showed a significant difference in SUV max between STAS-positive (5.21) and STAS-negative patients (2.42). However, MTV and TLG were not found to be useful in predicting the presence of STAS in stage I lung ADC. The relationship between STAS and PET parameters was further assessed based on nodule morphology. While no significant differences were observed in solid nodules between PET features and STAS, part-solid nodules exhibited a significant difference in SUV max: 3.41 in STAS-positive cases versus 1.60 in STAS-negative cases. In their methodology, patients fasted for at least six hours prior to undergoing PET/CT scans. FDG was administered intravenously, followed by a 60-min waiting period before scanning from the head to the upper thigh. The study concluded that tumor volume may not adequately reflect the risk of STAS. Instead, STAS-positive expression was primarily associated with increased FDG activity, emphasizing the role of SUV max as a potential predictor [10].

#### 3.3.2. Computed Tomography

The total size of the lesion observed in diagnostic imaging appears to be significantly associated with the presence of STAS. Tokoyawa et al. in their analysis of lung ADC reported a notable difference in STAS prevalence between lesions larger than 2 cm (54% of STAS-positive) and those smaller than 2 cm (42% of STAS-positive) [37]. Their study also demonstrated a strong correlation between a higher consolidation-to-tumor (C/T) ratio and the presence of STAS. In a univariable analysis, factors such as the presence of a notch and the absence of ground glass opacity (GGO) were significantly associated with STAS. In contrast, their multivariable analysis revealed additional significant predictors of STAS-positive AC, including vascular convergence, pleural indentation, the presence of a notch, spiculation, and a tumor diameter larger than 2 cm, compared to STAS-negative cases [37]. Research conducted by De Margerie-Mellon et al. found that STAS was associated with lesions larger than 2 cm and with subsolid lesions containing solid components ≥ 10 mm [36]. STAS-positive nodules exhibited significantly greater average and long-axis diameters, as well as a higher proportion of solid component diameter relative to the overall average diameter, compared to STAS-negative nodules [36].

The presence of STAS appears to be strongly associated with solid nodule type. Both Shiono et al. and Kim et al. reported similar findings in their studies. Shiono et al. observed that 79% of STAS-positive lesions were solid, while 21% were non-solid [38]. Similarly, Kim et al. found that 77% of STAS-positive lesions were solid, with 23% being part-solid [39]. The study by Kim et al. revealed that the proportion of the solid component (PSC), was significantly associated with STAS, with a proposed optimal threshold of 90%, according to multivariable analysis, which demonstrated good sensitivity [39]. Interestingly, only 36.4% of the STAS-negative nodules in this study presented as pure solid lesions. Additionally, the absence of an air bronchogram, the presence of poorly defined peripheral opacities, and central low attenuation were also identified as features associated with STAS [39]. In a study on invasive mucinous ADC (IMA), Lee et al. identified a statistically significant correlation between the presence of STAS and specific CT imaging characteristics, including lobulated margins (*p* = 0.006) and spiculated margins (*p* = 0.027) [31]. These features were thought to reflect the tumor’s interaction with the surrounding healthy lung parenchyma. Additionally, their analysis revealed that older age was significantly associated with IMA patients who had STAS [31]. Chen et al. sought to develop a nomogram to predict the STAS status of stage IA lung using CT imaging, achieving high predictive efficiency [40]. Nineteen morphological traits were analyzed, revealing that STAS-positive nodules were more likely to be solid, exhibit lobulation, have a larger solid component diameter, and a higher proportion of the solid component (PSC). The final nomogram for estimating STAS potential incorporated PSC and nodulation, identified in multivariable analysis as independent risk factors for STAS, with odds ratios (ORs) of 11.27 and 3.05, respectively [40].

### 3.4. Chemotherapy

Platinum-based chemotherapy is the preferred adjuvant treatment for patients with resectable stage IB-IIIA NSCLC and is considered the most effective component of the available treatment options [1]. However, the use of chemotherapy in patients with STAS remains a topic with limited data. Chen et al. reported that adjuvant chemotherapy (ACT) improved outcomes (overall survival, disease free survival) in patients with stage IB ADC with STAS, and in those with stage IA ADC/STAS who underwent sublobar resection [34]. For patients with stage IB ADC/STAS-positive, ACT was revealed as an independent factor for favorable. Among patients with stage IA ADC/STAS-positive, ACT was associated with improved outcomes only for those undergoing sublobar resection [34]. According to Yilv Lv et al. the necessity of ACT for stage I lung ADC remains controversial [41]. ACT did not improve survival in stage IA patients with STAS when administered postoperatively. However, stage IB patients with high-risk recurrence factors, ACT significantly improved 5-year recurrence-free survival (RFS) compared to those who did not receive it.

High-risk factors in stage IB include poorly differentiated tumors, lymphovascular invasion, and visceral pleural invasion, making such patients candidates for platinum-based ACT.

The decision to administer adjuvant chemotherapy is ultimately guided by oncologists’ assessment and patient preferences [41].

### 3.5. Radiotherapy

The management of NSCLC varies depending on the stage of the disease. Stereotactic body radiation therapy (SBRT), a high-dose targeted radiation treatment, is recommended for patients with comorbidities that preclude them from undergoing major surgical procedures. For locally advanced NSCLC, resectability plays a key role in determining treatment stratification. Patients with unresectable disease require definitive chemoradiation therapy, while those with resectable disease may benefit from additional treatments, such as chemotherapy and radiation [42].

Reports on the use of radiotherapy in the context of STAS remain limited. K. Makita et al. described the application of SBRT in early stage lung cancer with STAS involvement [43]. In their study, a peripheral lung tumor was treated with a dose of 48 Gy, while a centrally located tumor was treated with 60 Gy. However, applying the STAS concept to radiotherapy-poses challenges, as STAS is typically diagnosed through postoperative tissue specimens. K. Makita et al. demonstrated that patients receiving SBRT for early stage lung cancer have showed a correlation between the pretreatment status of Tumor-Cell Invasion through Air Spaces (TCIAS) and key survival metrics, including recurrence-free survival (RFS), distant failure-free survival (DFF), and progression-free survival (PFS) [43]. The TCIAS status may serve as a valuable prognostic marker for predicting SBRT outcomes in these patients. Nevertheless, further research, including large-scale studies with extended follow-up, is necessary to validate these findings [43].

### 3.6. Immunotherapy

First-line platinum-based doublet chemotherapy and second-line docetaxel have traditionally been the cornerstone treatments for patients with advanced-stage NSCLC who maintain good performance status. A novel therapeutic approach, immunotherapy, explores the potential of various vaccines and checkpoint inhibitors, that may offer significant long-term benefits [17]. Ongoing research is exploring the relationship between STAS and various immunotherapy targets. Kadota et al. analyzed recent data suggesting a correlation between anaplastic lymphoma kinase (ALK) rearrangements and specific histological features, such as the cribriform pattern in lung ADC, to assess the potential relevance of ALK gene mutation [33]. Other studies have found no significant correlation between STAS and programmed death-ligand 1 (PD-L1), v-Raf murine sarcoma viral oncogene homolog B1 (BRAF), human epidermal growth factor receptor 2 (HER2), or epidermal growth factor receptor (EGFR). As research into the molecular biological characteristics of STAS progresses, additional insights are expected to emerge [18,19,44].

### 3.7. Surgical Treatment

Lobectomy is the recommended surgical procedure for patients with early stage NSCLC. The JCOG0802/WJOG4607L trial is the first randomized study to demonstrate the superiority of segmentectomy over lobectomy in terms of overall survival for patients with early stage lung cancer. The findings suggest that segmentectomy should be considered the standard surgical approach for patients with clinical stage IA, small (≤2 cm, consolidation/tumor ratio > 0.5) peripheral NSCLC, rather than lobectomy [45]. However, it is important that this study did not account for the presence of STAS. There is considerable variation in prognostic outcomes among patients with NSCLC at the same stage of disease [32]. STAS has been established as an adverse prognostic factor in stage I lung ADC [3,32,46]. A meta-analysis by Yang et al. demonstrated that among patients with stage I lung ADC, the 5-year relapse-free survival (RFS) and overall survival (OS) rates were significantly lower in the STAS-positive group compared to the STAS-negative group [20]. In stage IA disease, patients with STAS had postoperative outcomes similar to those with stage IB and inferior outcomes compared to those without STAS [46]. Additionally, STAS has been identified as an independent predictor of recurrence and lung cancer-specific death in SCC [21]. Among patients with stage I SCC, STAS-positive cases exhibited significantly lower 5-year RFS and OS rates than STAS-negative cases. However, this correlation was not observed in stages II and III. A multivariate analysis confirmed that STAS is an independent prognostic factor and predictor of recurrence in stage I squamous cell carcinoma but not in stages II or III [8].

The presence of STAS is a critical factor that surgeons must evaluate when determining the extent of surgical intervention [32]. A study by Lv et al. found that among patients with stage IA lung ADC exhibiting STAS, those who underwent lobectomy had a prognosis comparable to patients treated with sublobar resection [41]. Similarly, Kagimoto et al. reported that for lung adenocarcinoma patients with STAS, segmentectomy outcomes were comparable to those of lobectomy, with no observed increase in locoregional recurrence in stage IA [5]. The impact of STAS on recurrence and mortality rates of lung ADC appears to be significantly reduced by lobectomy in comparison to sublobar resection among patients diagnosed with stage I cancer [7,47]. Studies comparing lobectomy and sublobectomy in stage I lung cancer with STAS, including some with stage IB disease, suggest that sublobar resection may be less effective. Most recurrences in STAS-positive patients who underwent sublobar resection were locoregional, highlighting that a wider resection margin may not adequately prevent recurrence in these cases [7]. This conclusion aligns with a meta-analysis by Yang et al., which found that sublobectomy is associated with a higher risk of recurrence and worse 5-year survival rates in STAS-positive patients, with an 5 year RFS HR = 6.92, 95% CI (1.64–12.18) [20]. Yanagawa et al. investigated lung SCC across all stages and found that patients with STAS had significantly worse 5-year RFS and 5-year OS compared to those without STAS within the sublobectomy cohort (*p* = 0.0036 and *p* = 0.035, respectively) [8]. In the lobectomy group, patients with STAS also showed a tendency toward poorer 5-year RFS and 5-year OS compared to their STAS-negative counterparts, though the differences did not reach statistical significance (*p* = 0.056 and *p* = 0.111, respectively). Among stage I patients, those with STAS had significantly inferior 5-year RFS and 5-year OS compared to STAS-negative patients in the sublobar resection group (*p* = 0.0036 and *p* = 0.033, respectively). Similarly, in the lobectomy group, patients with STAS experienced worse 5-year RFS, with a trend toward decreased 5-year OS, though the latter was not statistically significant (*p* = 0.058) [8].

## 4. Discussion

Y. Meng et al. identified sex, E-cadherin, N-cadherin, and FAK as independent predictors of STAS [22]. These adhesion molecules facilitate STAS dissemination through the following mechanisms: upon phosphorylation, FAK—a key tyrosine kinase in the integrin signaling cascade—assembles into an FAK/Src complex that orchestrates actin cytoskeletal remodeling by phosphorylation, inducing focal adhesion relaxation and thereby modulating adhesion dynamics, cytoskeletal reorganization, and cell migration. Concurrently, aberrant activation of epithelial–mesenchymal transition (EMT) enhances cell motility, invasiveness, and apoptotic resistance. During EMT, the downregulation of epithelial E-cadherin and upregulation of mesenchymal N-cadherin replace rigid intercellular adhesions with more flexible junctions, promoting detachment and migration. Moreover, univariate analysis revealed that the expression levels of E-cadherin, N-cadherin, and FAK correlate significantly with tumor growth patterns and STAS. After adjusting for age, surgical procedure, and visceral pleural invasion, multivariate analysis further demonstrated that sex and the N-cadherin/FAK risk index independently predict recurrence-free probability in STAS-positive patient [22].

Another independent factor connected with high STAS incidents is matrix metalloproteinase known as matrilysin (MMP-7) [23]. From a biochemical point of view MMP-7 is zinc-dependent endopeptidase, which presents proteolytic activity against components of the extracellular matrix (ECM), such as elastin, type IV collagen, fibronectin, vitronectin, aggrecan, and proteoglycans [24]. In addition, MMP-7 contributes to innate immune responses in the lung and intestine and is involved in the proteolytic shedding of cell-surface molecules. What is significant, expression of MMP-7 is enhanced in various epithelial and mesenchymal tumors. Moreover, MMP-7 is engaged in each carcinogenesis phase, from initiation to creating metastasis. Rasheed et al. provided evidence that suppression of MM-7 leads to inhibition of invasion and metastasis in NSCLC [25]. According to Yamada et al.’s own interpretation, the process begins with MMP-7 degrading various matrix substrates; subsequently, tumor STAS cells detach from the primary tumor, migrate through the air spaces beyond the tumor margin, and ultimately reattach to the alveolar walls leading to intrapulmonary spread [23].

Tumor-associated fibroblasts (CAFs) and tumor-associated macrophages (TAMs) represent significant components of the tumor microenvironment, with the potential to influence cancer dissemination and metastasis. Studies by Xie Qiu et al. indicate the association between stromal cells and the presence of STAS in NSCLC and suggested the potential for using these associations to estimate the prognosis of patients [26]. Immunohistochemical methods were used to assess the expression of α-smooth muscle actin (SMA) markers—positive CAFs and CD204-positive TAMs. Research has demonstrated a correlation between the presence of STAS and elevated levels of α-SMA and CD204 in the context of lung adenocarcinoma. Furthermore, the presence of STAS and the high frequency of α-SMA-positive CAFs and CD204-positive TAMs were statistically significant independent predictors of OS and RFS. Patients demonstrating a high incidence of α-SMA-positive CAFs exhibited inferior OS and RFS in comparison to patients exhibiting low expression levels across both the STAS-positive and STAS-negative groups. However, it was observed that only patients in the STAS-positive group with high CD204 expression had worse OS and recurrence-free survival RFS than patients with low CD204 expression. α-SMA-positive CAFs play an important prognostic role in patients with postoperative lung adenocarcinoma, regardless of the presence of STAS. However, CD204-positive TAMs significantly predicted prognosis only in the presence of STAS [26].

NSCLC is associated with a variety of biomarkers that not only reflect the tumor’s biological behavior and stage but may also serve as prognostic indicators of patient outcomes. In the discussion below, we focus briefly on the biomarkers and molecular alterations observed in STAS.

Vascular endothelial growth factor (VEGF) is a proangiogenic peptide that stimulates the growth and development of vascular endothelial cells, stimulates proliferation and differentiation, prolongs the viability of existing vessels, and thus promotes tumor growth [27]. Local hypoxia is the main inducer of VEGF gene expression in the tumor microenvironment [28].

VEGF expression has been observed to be excessive in non-small cell lung cancer (NSCLC). Elevated levels of VEGF have been demonstrated to be associated with tumor recurrence, reduced survival rates, metastasis, and death [29,48]. In a study on a group of 86 patients diagnosed with NSCLC, Naikoo et al. evaluated the expression of VEGF in resected tumor tissue and distant treatment outcomes [27]. The study found that higher VEGF expression was an independent negative prognostic factor. However, no correlation was observed between VEGF expression and histopathological diagnosis or the stage of lung cancer [27]. VEGF is essential for tumor growth and immunosuppression. Consequently, targeted drugs that inhibit the VEGF pathway are employed in the treatment of non-small cell lung cancer [29,48].

Guo et al. found that higher Vascular Endothelial Growth Factor A (VEGFA) expression, mediated by hypoxia-inducible factor 1 (HIF-1) signaling, was associated with an increased STAS rate [49].

HIFs plays a significant role in the regulation of various genetic mechanisms responsible for the survival of tumor cells and is a modulator of immune response by affecting natural killer cell-mediated antitumor response [50,51]. HIF-1α expression was associated with tumor proliferation, resistance to apoptosis, and increased mortality in patients with lung cancer, which indicates its potential prognostic significance [52,53]. YKL-40 protein is implicated in the proliferation and differentiation of tumor cells and protects them from apoptosis. It plays a critical role in angiogenesis, facilitates the reconstruction of the extracellular matrix, and stimulates fibroblasts within the tumor microenvironment [54,55]. Association between serum YKL-40 levels and patient survival imply that elevated YKL-40 may serve as a promising prognostic indicator in advanced non-small cell lung cancer (NSCLC) [54,55,56,57,58]. In addition to the aforementioned tumor indicators, serum markers include carcinoembryonic antigen (CEA), nerve-specific enolase (NSE), cytokeratin 19 fragment (CYFRA 21-1), squamous cell carcinoma antigen (SCC-Ag), and pro-gastrin-releasing peptide (ProGRP) [59]. The levels of these markers were significantly higher in the serum of lung cancer patients compared to the healthy control group. When these indicators are evaluated together, their diagnostic value is greater than when only one biomarker is assessed [59].

Only vascular endothelial growth factor (VEGF) has been unequivocally linked to the enhancement of STAS. Although the other biomarkers have demonstrated significant utility in the diagnosis, staging, and prognostication of NSCLC, none have yet been shown to correlate directly with STAS. Therefore, it is imperative that future studies investigate their potential roles in detecting STAS, determining optimal resection margins, and elucidating their prognostic significance.

STAS remains an emerging topic, and reports regarding its radiological diagnosis, chemotherapy, radiotherapy, and the extent of surgical intervention are still limited. Consequently, it is challenging to compare the information we have presented from scientific publications with other data; such comparisons will only become feasible in the future as this issue is explored more widely. In our review, we included all studies on STAS indexed in PubMed, published since 2015, and focused exclusively on adult populations. Four authors independently conducted the review, searching only for English-language databases, which may have led to the omission of potentially relevant studies. We did not perform a statistical meta-analysis due to the small number of heterogeneous studies addressing our topics of interest, leading to few study participants and the absence of uniform validation criteria, which may limit the precision of our conclusions. Lung cancer, as demonstrated earlier, represents a significant challenge for oncological patients; therefore, a precise understanding of STAS is crucial. For this reason, we endeavored to present key clinical and therapeutic aspects concerning STAS. We highlighted the histological subtypes in which STAS cells occur most frequently, discussed various imaging techniques for assessing the risk of STAS preoperatively, and reviewed the available evidence on the efficacy of chemotherapeutic and radiotherapeutic regimens. One of the focal points of our work was the scope of surgical procedures carried out in cases of STAS diagnosis, their extent, and the subsequent impact on patient prognosis and recurrence risk. We also addressed the biology of STAS and its potential mechanisms of spread, noting, however, the lack of studies measuring biomarker concentrations in published research, an area that should be prioritized in future investigations.

## 5. Conclusions

The scientific research on STAS remains limited. In patients with NSCLC, the presence of STAS is recognized as a negative prognostic factor. Assessment of STAS in patients with tumors smaller than 3 cm is necessary to decide on the extent of tumor resection. Due to the fact that the current gold standard in the assessment of this parameter is a postoperative pathomorphological examination, methods are sought to be able to assess in advance the presence or absence of STAS. Emerging studies increasingly suggest that imaging techniques may play a valuable role in its detection. However, there is still insufficient evidence to guide specific treatment strategies when STAS is identified. Further research is therefore needed to establish more precise treatment protocols and improve patient outcomes.

## Figures and Tables

**Figure 1 cancers-17-01696-f001:**
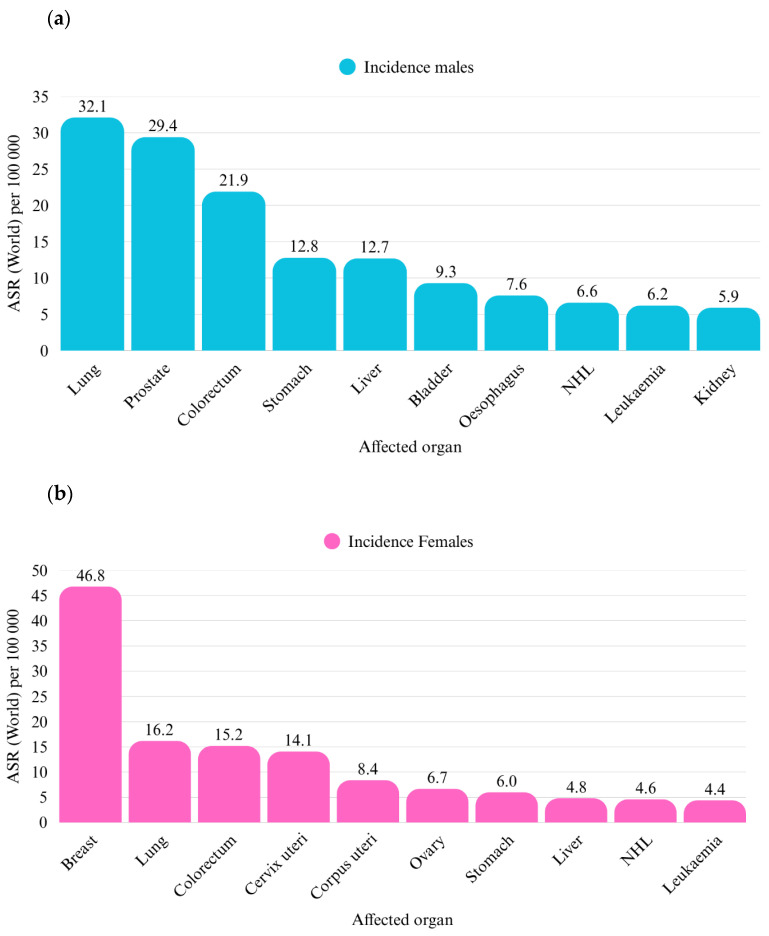
Age-standardized cancer incidence rate per 100,000 persons worldwide, divided into men (**a**) and women (**b**) in 2022. The data used to generate the chart were sourced from the International Agency for Research on Cancer (IARC), Global Cancer Observatory (IARC, 2023) [1].

**Figure 2 cancers-17-01696-f002:**
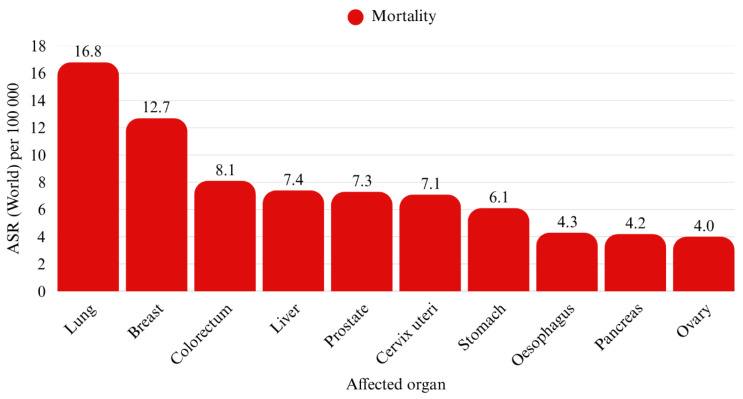
Age-standardized cancer mortality rate per 100,000 persons worldwide in 2022. The data used to generate the chart were sourced from the International Agency for Research on Cancer (IARC), Global Cancer Observatory (IARC, 2023) [1].

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
