# Peer review of "Spread Through Air Spaces (STAS) as a Predictive and Prognostic Factor in Patients with Non-Small Cell Lung Cancer—Systematic Review"

_cancers, 2025, doi:10.3390/cancers17101696_

Round 1

Reviewer 1 Report

Comments and Suggestions for Authors

"Spread through air spaces (STAS) as a predictive and prognostic factor in patients with non-small cell lung cancer" is an important topic and have potential applications but there are following concerns that need to be addressed before publication:

  1. Authors should highlight the points of innovation in the abstract.
  2. Authors should add schematics and graphical figures to increase the effectiveness and readership of the study.
  3.  Authors should add some real-life examples in each section.
  4. Authors should add some real research diagrams from published literature with due copyright.
  5. Authors should elaborate the conclusions section and also discuss the future perspectives.
  6. Authors should cite the recent literature.

Author Response

Dear Sirs

Thank you for your comments on our work. Below we send you our position on the topics you raised.

Comments 1: Authors should highlight the points of innovation in the abstract.

Response 1: We emphasized the innovative nature of this work in the discussion. Page 15, Lines 456-459, 467-474.

STAS remains an emerging topic, and reports regarding its radiological diagnosis, chemotherapy, radiotherapy, and the extent of surgical intervention are still limited. Consequently, it is challenging to compare the information we have presented from scientific publications with other data; such comparisons will only become feasible in the future as this issue is explored more widely.”

“. Lung cancer, as demonstrated earlier, represents a significant challenge for oncological patients; therefore, a precise understanding of  STAS is crucial. For this reason, we endeavored to present key clinical and therapeutic aspects concerning STAS. We highlighted the histological subtypes in which STAS cells occur most frequently, discussed various imaging techniques for assessing the risk of STAS preoperatively, and reviewed the available evidence on the efficacy of chemotherapeutic and radiotherapeutic regimens. One of the focal points of our work was the scope of surgical procedures carried out in cases of STAS diagnosis, their extent, and the subsequent impact on patient prognosis and recurrence risk. We also addressed the biology of STAS and its potential mechanisms of spread, noting, however, the lack of studies measuring biomarker concentrations in published research, an area that should be prioritized in future investigations.”

Comments 2: Authors should add schematics and graphical figures to increase the effectiveness and readership of the study.

Response 2: We decided to improve the graphs at the beginning of our work regarding the incidence and mortality of individual cancers in order to emphasize the importance of the problem of lung cancer. – Page 3 and 4 – Figures 1 and 2

Comments 3: Authors should add some real-life examples in each section.

Response 3:

Thank you for this feedback. In each section, we have tried to show the impact that the presence of STAS can have as a negative prognostic and predictive factor for a given treatment modality. Currently, the best described example of the practical use of stas knowledge is the decision between lobectomy and sublobar resection. Which has been mentioned in several places in our work: Page 5 lines 115-119, Page 12 Lines 363-367, Page 12, lines 369-371

Comments 4: Authors should add some real research diagrams from published literature with due copyright.

Response 4:  We replaced the graphs in the first version of our work with graphs we created based on the available literature. Page 3 and 4 – Figures 1 and 2

Comments 5: Authors should elaborate the conclusions section and also discuss the future perspectives.

Response 5:

Thank you for the suggestion; we have expanded the Conclusions. The current text now reads as follows: “The scientific research on STAS remains limited. In patients with NSCLC, the presence of STAS is recognized as a negative prognostic factor. Assessment of STAS in patients with tumors smaller than 3 cm is necessary to decide on the extent of tumor resection. Due to the fact that the current gold standard in the assessment of this parameter is a postoperative pathomorphological examination, methods are sought to be able to assess in advance the presence or absence of STAS. Emerging studies increasingly suggest that imaging techniques may play a valuable role in its detection. However, there is still insufficient evidence to guide specific treatment strategies when STAS is identified. Further research is therefore needed to establish more precise treatment protocols and improve patient outcomes.

More research is needed on STAS to be able to more precisely define recommendations regarding treatment options for patients with a positive result for this parameter.”

Page 15, lines 477-485

Comments 6: Authors should cite the recent literature.

Response 6: Thank you for your response. In our manuscript, the vast majority of cited studies were published between 2017 and 2025. Only six references date from before 2010; these were included because they address lung-cancer biomarkers rather than STAS specifically. We hope this clarifies any concerns regarding the publication dates of our sources. Should you have any further questions, we would appreciate your additional suggestions.

Reviewer 2 Report

Comments and Suggestions for Authors

Half the introduction is a mini-review on VEGF, HIF-1α, YKL-40, and general serum markers. The tangent dilutes focus and never comes back to why those sections matter for a STAS-centric paper.

Never define which outcomes are being “predicted” (treatment benefit? recurrence pattern?) versus merely “prognosed.”

Study types are named but selection thresholds (sample size, histology confirmation, STAS definition) are absent. Risk-of-bias tools are not applied.

Prognostic value of STAS under lobectomy vs. sublobar surgery is reiterated from prior meta-analyses, but you omit salvage factors.

No multivariate model, no calibration, no discrimination metrics (AUC/C-index), and no external validation are provided.

Author Response

Dear Sirs

Thank you for your comments on our work. Below we send you our position on the topics you raised.

Comment 1: Half the introduction is a mini-review on VEGF, HIF-1α, YKL-40, and general serum markers. The tangent dilutes focus and never comes back to why those sections matter for a STAS-centric paper.

Response 1: Thank you for your review. We fully agree with your opinion and your feedback has been taken into account. References to serum markers related to lung cancer have been kept to minimum, serving only to provide necessary context for the readers without causing distraction.

Comment 2: Never define which outcomes are being “predicted” (treatment benefit? recurrence pattern?) versus merely “prognosed.”

Response 2: Thank you for this thoughtful comment. In our work, we adopted the distinction between prognostic and predictive outcomes based on established definitions in the literature, as discussed in our literature review. While we understand the importance of clearly differentiating these terms, we believe that our current usage accurately reflects the framework supported by the sources.

Comment 3: Study types are named but selection thresholds (sample size, histology confirmation, STAS definition) are absent. Risk-of-bias tools are not applied.”

Response 3: Thank you for your comment. Regarding the STAS definition, we followed the criteria used in the studies included in our review and provided the adopted definition clearly in the Intoduction. For histological confirmation, we dedicated a separate paragraph to discussing the presence and interpretation of STAS across different histological subtypes of lung cancer, reflecting how this was addressed in the original studies. While we did not apply formal risk-of-bias tools our aim was to provided a narrative synthesis of existing evidence rather than a systematic quality appraisal. We did report sample size when it was relevant to the interpretation of findings or highlighted by the original study. 

Comment 4: Prognostic value of STAS under lobectomy vs. sublobar surgery is reiterated from prior meta-analyses, but you omit salvage factors.No multivariate model, no calibration, no discrimination metrics (AUC/C-index), and no external validation are provided.

Response 4: Thank you for this comment. In this section, our aim was to summarize the current evidence regarding the intraoperative evaluation of STAS and its role in guiding surgical decisions, particularly between lobe Tony and sublobar resection. We presented key findings from primary studies that included multi variable analyses (e.g., Eguchi et al.), as well as sensitivity and specificity data for frozen section assessment. As our review does not propose or test new prognostic models, we did not include model performance metrics such as calibration, discrimination or external validation. Regarding salvage factors, we acknowledge their potential relevance but considered them beyond the scope of this specific discussion focused on intraoperative decision-making.

Reviewer 3 Report

Comments and Suggestions for Authors

This manuscript reviews current understanding of the spread through air spaces of lung cancer cells as a parameter for prediction and prognosis in the development of non-small cell lung cancer. The topic is appropriately focussed, and a collection of relevant and impactful publications has been summarised and key findings presented clearly. Some insight has been given into the critical areas of future research in this field. The manuscript is mostly clearly written and easy to read. There are some points that the authors should address, and these are detailed below (all points needing changes to the manuscript).

  1. Page 1 lines 29-30 “Platinum-based adjuvant . . . with STAS . . .” This text does not quite make sense. What aspect of resectable NSCLC with STAS does platinum-based adjuvant chemotherapy and SBRT show promise for? Diagnosis, prognosis estimation, predictive potential?
  2. I note that figures 1 and 2 omit breast cancer, which at first glance seems odd considering that breast cancer is the most common invasive cancer in women in most countries of the world. Some clarification might be merited here.
  3. Page 4 lines 102-104 “In hypoxic circumstances . . . a transcription factor.” This description as written suggests that under hypoxic conditions, the HIF-1α cannot migrate to the nucleus but then go on to talk of the complex being capable of reacting with DNA and that this is implicated in the ultimate survival of tumour cells. This seems contradictory. Can you please clarify the wording?
  4. Page 4 line 145 – page 5 line 148 “In addition to . . . (ProGRP).” This statement needs a supporting reference/s.
  5. Your referencing system uses a numerical list and in-text numerical citation. Therefore, when you use the form ‘Huijuan et al.’ or similar, then you still need to add the numerical citation so that the reference can be found more easily from the reference list. This applies in multiple places throughout the manuscript.
  6. Page 7 lines 245-246 “. . . to assess the potential . . . another gene mutation.” Which gene mutation? Some extra succinct detail on the purpose of this work is needed to clarify the relevance of this reference to the topic of the review.
  7. Page 7 line 287 “. . . RFS HR = 6.92, 95% CI (1.64-12.18)45.” You need to add units here (presumably years). This applies both here and elsewhere in the manuscript.
  8. Page 10 lines 411 and 412 It is not clear why P = 0.006 and P = 0.027 have been given three times each in the text. If this is a mistake then please correct it, otherwise clarify the reason for it.

Comments on the Quality of English Language

The manuscript could benefit from a light editorial check on grammar and syntax.

Author Response

Thank you for your comments on our work. Below we send you our position on the topics you raised. We kindly request further guidance

Comments 1: Page 1 lines 29-30

"Platinum-based adjuvant . . . with STAS . . ."
This text does not quite make sense. What aspect of resectable NSCLC with STAS does platinum-based adjuvant chemotherapy and SBRT show promise for? Diagnosis, prognosis estimation, predictive potential?
Response 1: Thank you for your suggestion. The abstract has been revised accordingly. Additionally, we have amended the paragraph discussing chemotherapy to specify that chemotherapy improves both overall survival (OS) and disease-free survival (DFS) (see page 10, lines 285–290). If these revisions are deemed insufficient or have been misinterpreted, we kindly request further guidance. “Chen et al. reported that adjuvant chemotherapy (ACT) improved outcomes  (overall survival , disease free survival) in patients with stage IB ADC with STAS,  and in those with stage IA ADC/STAS who underwent sublobar resection.19 For patients with stage IB ADC/STAS-positive, ACT was revealed as an independent factor for favorable. Among patients with stage IA ADC/STAS-positive, ACT was associated with improved outcomes only for those undergoing sublobar recestion. 19 According to Yilv Lv et al. the necessity of ACT for stage I lung ADC remains controversial.30 ACT did not improve survival in stage IA patients with STAS when administered postoperatively. However, stage IB patients with high-risk recurrence factors, ACT significantly improved 5-year recurrence-free survival (RFS) compared to those who did not receive it.”

Comments 2: I note that figures 1 and 2 omit breast cancer, which at first glance seems odd considering that breast cancer is the most common invasive cancer in women in most countries of the world. Some clarification might be merited here.
Response 2: Thank you for pointing this out. The chart did not contain information on breast cancer. We decided to generate the charts ourselves and also supplemented it with the most important cancers in women (including breast cancer) – Page 3, Figure 1.

Comments 3: Page 4 lines 102-104
"In hypoxic circumstances . . . a transcription factor."
This description as written suggests that under hypoxic conditions, the HIF-1α cannot migrate to the nucleus but then go on to talk of the complex being capable of reacting with DNA and that this is implicated in the ultimate survival of tumour cells. This seems contradictory. Can you please clarify the wording?
Response 3: Following suggestions from another reviewer, we have decided to remove this section describing HIF-1α.

Comments 4: Page 4 line 145 – page 5 line 148
"In addition to . . . (ProGRP)."
This statement needs a supporting reference/s.
Response 4: Agree. We have added a reference – Page 14, line 447.

Comments 5: Your referencing system uses a numerical list and in-text numerical citation. Therefore, when you use the form ‘Huijuan et al.’ or similar, then you still need to add the numerical citation so that the reference can be found more easily from the reference list. This applies in multiple places throughout the manuscript.
Response 5: After each citation of the source in the form of "Author et al.", we placed a citation at the end of the sentence.

Comments 6: Page 7 lines 245-246
"…to assess the potential . . . another gene mutation."
Which gene mutation? Some extra succinct detail on the purpose of this work is needed to clarify the relevance of this reference to the topic of the review.

Response 6: We added the name of the mutated gene.  Page 11, line 326.

Comments 7: Page 7 line 287
"...RFS HR = 6.92, 95% CI (1.64-12.18)⁴⁵."
You need to add units here (presumably years). This applies both here and elsewhere in the manuscript.
Response 7: Agree. We have added the units (years) for both OR and RFS – Page 12, lines 362-372.

Comments 8: Page 10 lines 411 and 412
It is not clear why P = 0.006 and P = 0.027 have been given three times each in the text. If this is a mistake then please correct it, otherwise clarify the reason for it.
Response 8: There was an error in the quoted lines, which we have corrected. Page 9, lines 271-273.

Round 2

Reviewer 2 Report

Comments and Suggestions for Authors

The authors answered all my questions. The manuscript has been sufficiently improved to warrant publication.